# Dandelion-Like CuCo_2_O_4_@ NiMn LDH Core/Shell Nanoflowers for Excellent Battery-Type Supercapacitor

**DOI:** 10.3390/nano13040730

**Published:** 2023-02-14

**Authors:** Wenhua Zhao, Xingliang Xu, Niandu Wu, Xiaodie Zhao, Jiangfeng Gong

**Affiliations:** 1Department of Applied Physics, Zhejiang University of Science and Technology, Hangzhou 310023, China; 2National Laboratory of Solid-State Microstructures, Collaborative Innovation Center of Advanced Microstructures, School of Physics, Nanjing University, Nanjing 210093, China; 3College of Science, Hohai University, Nanjing 211199, China

**Keywords:** CuCo_2_O_4_@NiMn LDH nanoflowers, core/shell, synergistic effect, battery-type, supercapacitors

## Abstract

Dandelion-like CuCo_2_O_4_ nanoflowers (CCO NFs) with ultrathin NiMn layered double hydroxide (LDH) shells were fabricated via a two-step hydrothermal method. The prepared CuCo_2_O_4_@NiMn LDH core/shell nanoflowers (CCO@NM LDH NFs) possessed a high specific surface area (~181 m^2^·g^−1^) with an average pore size of ~256 nm. Herein, the CCO@NM LDH NFs exhibited the typical battery-type electrode material with a specific capacity of 2156.53 F·g^−1^ at a current density of 1 A·g^−1^. With the increase in current density, the rate capability retention was 68.3% at a current density of 10 A·g^−1^. In particular, the 94.6% capacity of CCO@NM LDH NFs remains after 2500 cycles at 5 A·g^−1^. An asymmetric supercapacitor (ASC) with CCO@NM LDH NFs//activated carbon (AC) demonstrates a remarkable capacitance of 303.11 F·g^−1^ at 1 A·g^−1^ with excellent cycling stability. The coupling and synergistic effects of multi-valence transition metals provide a convenient channel for the electrochemical process, which is beneficial to spread widely within the realm of electrochemical energy storage.

## 1. Introduction

Supercapacitors have attracted extensive research attention due to the increasing demand for sustainable and reliable energy storage devices. Simultaneously, supercapacitors can bridge the gap between batteries and capacitors since they can provide power density and long cycle life [1,2]. Binary transition metal oxides have been considered to be promising candidates as electrodes for high-performance pseudo-capacitors due to their multiple redox reactions and high electrical conductivity [3,4,5]. Because of the various redox reactions and high conductivity, transition bimetallic oxides are regarded as the best candidates for high-performance pseudocapacitor electrode materials [6,7].

The electrode material with core/shell structures can provide more active sites to improve its conductivity, achieving unprecedented electrochemical performance. The main reason for this is that the core/shell structure can promote the rapid transfer of ions and electrons in the electrochemical process [8]. Moreover, the electrode material in the core possesses high conductivity, which can optimize the transmission path of charge. The shell with a higher surface area can further promote the mass transfer at the interface of the electrode and electrolyte, further providing more space for ion transport. In addition, more defects [9,10], such as oxygen vacancies, can be introduced using core/shell structures, which can alleviate the change in sample volume in the electrochemical test. In recent years, more and more efforts have been made to construct core/shell electrode materials, such as MnCo_2_O_4_@Ni(OH)_2_ core-shell flowers [11], (Ni,Co)Se_2_/NiCo-LDH core/shell [12], P-Ni(OH)_2_@Co(OH)_2_ core/shell [13], Mn(OH)_2_-containing Co(OH)_2_/Ni(OH)_2_ core/shell [14], MoS_2_@α-Fe_2_O_3_/CNTF [15], MnS@Ni(OH)_2_ core/shell [16], Ni(OH)_2_@NHCSs electrode [17], NiCoP@C@Ni(OH)_2_ core/shell nanoarrays [18], and so on. The rational construction of the core/shell electrode greatly promotes the development of supercapacitors through a synergetic effect. Two-dimensional NiMn-layered hydroxide (NiMn LDH) with low cost demonstrates multiple valence states of transition metals, such as Ni and Mn, which can facilitate rich redox reactions in the electrochemical process [19,20]. At the same time, NiMn LDH increases the conductivity of electrode materials and accelerates ion transmission [21,22].

Herein, a CCO core is shelled with NiMn LDH, which forms CCO@NM LDH NFs’ electrode material via two hydrothermal reactions. CCO@NM LDH NFs with core/shell structure can construct abundant oxygen vacancy defects to increase the large specific surface area of CCO@NM LDH NFs, which enables ions to freely insert and remove in the electrochemical process, thus improving the supercapacitor performance of composite electrode materials. As the anode material of supercapacitors, the discharge-specific capacitance of core/shell CCO@NM LDH NFs is as high as 2156.53 F·g^−1^ at a current density of 1 A·g^−1^. Activated carbon and CCO@NM LDH NFs are used as the anode and cathode, respectively, for the construction of the asymmetric supercapacitor. The hybrid supercapacitor demonstrates a remarkable capacitance of 303.11 F·g^−1^ at 1 A·g^−1^. It also provides a maximum energy density of 92.2 Wh·kg^−1^ at a power density of 730 W·kg^−1^; moreover, and it shows excellent stability. Therefore, this work can be used as a guide for preparing new functional materials with high-performance supercapacitors.

## 2. Materials and Methods

### 2.1. Synthesis of Samples

The CCO@NM LDH is synthesized using a two-step hydrothermal reaction schematically shown in Figure 1. Firstly, 4 mmol of Co(NO_3_)_2_·6H_2_O and 2 mmol of Cu(NO_3_)_2_·6H_2_O were dissolved in 80 mL of distilled water and ethanol with a volume ratio of 1:1 as the starting materials. A piece of Ni foam (NF, 2 × 4 cm^2^) was submerged in the homogeneous solution and heated at 120 °C for 6 h using a hydrothermal reaction. Then, the CuCo precursor attached to NF surface was successfully obtained. the CuCo precursor powder was annealed at 350 °C for 6 h to obtain NF-loaded CuCo_2_O_4_. Secondly, Ni foam loaded with CuCo_2_O_4_ was used as a substrate for secondary hydrothermal growth in the following solution. The solution contains 0.713 g of NiCl_2_·6H_2_O and 0.198 g of MnCl_2_·4H_2_O, and 0.7 g of hexamethylenetetramine (HMT) and the NF-loaded CuCo_2_O_4_ were rinsed in 70 mL of distilled water. The reaction was conducted at 90 °C for 6 h. After the reaction, the mixture loaded on NF was washed and dried at 60 °C to obtain CCO@NM LDH NF samples.

### 2.2. Characterization

A field emission scanning electron microscope (FE-SEM, s4800, Hitachi, Tokyo, Japan) and transmission electron microscope (TEM, JEM-1200EX, Tokyo, Japan) were used to characterize the morphology of the samples. Energy dispersive X-ray spectroscopy (EDS) was used to display the distribution of elements. X-ray diffractometer (XRD, td-3500, Heilongjiang, China) in Cu K target (α = 1.542 Å) was recorded at the 2θ scanning range of 10° to 80° Kα. The chemical states of elements were studied using X-ray photoelectron spectroscopy (XPS, model, uivac phi, Tokyo, Japan). The specific surface area was analyzed via BET JW-004 with nitrogen adsorption/desorption for 1 h.

## 3. Results and Discussions

To better investigate the influence on the morphology of the dandelion-like CCO@NM LDH NFs, a series of FE-SEM images were carried out to determine the evolution of the CuCo precursors. Figure 1 shows that the morphologies of the CuCo precursor on Ni foam (NF) transformed from loosely assembled immature pine needle-like structures to densely stacked dandelion-like spheres after high-temperature annealing. Subsequently, ultrathin NiMn LDH layers were hydrothermally grown on the surface of dandelion-like CuCo_2_O_4_ nanoflowers, as illustrated in Figure 1a, which greatly increased the specific surface area of the CCO-NM LDH NFs, subsequently providing more active sites. The FE-SEM images of the CuCo_2_O_4_ sample and CuCo_2_O_4_-NiMn LDH sample are shown in Appendix A and Appendix A, respectively (see Appendix A). Dandelion-like CCO NFs are evenly dispersed on the surface of the NF with a size of about 1 μm. As shown in Figure 1b, dandelion-like CCO NFs are shelled with ultrathin NiMn LDH. The nanoflowers are interwoven, and their size increases to about 2 μm. The elemental distribution on CCO and CCO@NM LDH NFs was further investigated by performing energy dispersive (EDS) and elemental mapping in Appendix A and Figure 1c–i, respectively. Cu, Co, Ni, Mn, and O are homogeneously distributed on the entire surface of the CCO@NM LDH NFs, confirming the successful synthesis of a heterostructure in which NiMn LDH has grown on the surface of the CCO NFs.

The morphology and crystallinity of CuCo_2_O_4_-NiMn LDH material were further analyzed using high-resolution transmission electron microscopy (HRTEM) and XRD measurements. As shown in Figure 2a, the CCO@NM LDH material appears as a high-light transmission, indicating that the sample is also ultrathin with good dispersion and no obvious stacking. The size of the CCO@NM LDH material is as large as that shown in the SEM analysis. Notably, the clear and orderly lattice spacing of CCO-NM LDH NFs is 0.248 nm in Figure 2b which is, consistent with the (009) plane of NiMn LDH. From Figure 2c, two obvious diffraction rings of the selected area electron diffraction (SAED) also correspond to (003) and (009) planes of CCO@NM LDH material. The phase purity and crystal structure of the CCO and CCO@NM LDH NFs were investigated using XRD in Figure 2d. Several peaks are observed at 2θ values of 18.88, 31.08, 36.62, 38.30, 44.52, 58.96, and 64.79°, which are indexed to the (111), (220), (311), (222), (400), (511), and (440) crystal planes of cubic CuCo_2_O_4_ (JCPDS No. 78-2177). Then, the diffraction peaks of LDH can be reasonably indexed to a series of crystal planes, such as (003), (006), (009), and (110). This demonstrates the successful preparation of spinel CuCo_2_O_4_ shelled with NiMn LDH. No peaks of impurities such as copper oxides and cobalt oxides were found, confirming the high purity of the product. The synergistic effect of dandelion-like CuCo_2_O_4_ and ultrathin NiMn LDH resulted in the successful preparation of CCO@NM LDH NFs with good crystallinity.

As shown in Figure 3, XPS is used to identify the chemical state. The full XPS spectrum of CCO and CCO@NM LDH NFs is illustrated in Figure 3a. There is no interference from other impurity elements in the prepared samples. Figure 3b–d demonstrates XPS high-resolution spectra for Cu 2p, Co 2p, and O 1s, respectively. There are two sharp peaks at 952.6 eV and 932.6 eV, corresponding to Cu 2p_3/2_ and Cu 2p_1/2_, respectively [23]. The Co 2p spectrum in Figure 3c can be deconvoluted into peaks at 778.4 eV and 793.3 eV, in agreement with Co^3+^, while peaks at 780.1 eV and 794.8 eV can be corresponded to Co^2+^. Moreover, Co^2+^ and Co^3+^ co-exist in the CuCo_2_O_4_ sample and CCO@NM LDH NFs. It should be noted that the peak at 528.45 eV in the O1s spectrum disappears in the obtained CCO@NM LDH compared with CuCo_2_O_4_. The peak in the O1s spectrum of CuCo_2_O_4_ is fitted into three subpeaks, corresponding to O_L_ (528.4 eV), O_OH_ (529.9 eV), and O_V_ (531.1 eV). The O_L_ peak is ascribed to the Cu−O and Co−O bonds in Figure 3d. The O_OH_ peak is indexed to the −OH group [24]. O_L_ almost disappears with the synergistic effect of the core/shell structure. The appearance of O_W_, located at 532.4 eV, induced physically and chemically absorbed water molecules on the surface of CCO@NM LDH NFs [25]. Obviously, O_V_ increases due to the transformation of O_L_, and O_V_ represents oxygen vacancy. Meanwhile, peaks at 855.6 eV and 873.2 eV correspond to Ni 2p_3/2_ and Ni 2p_1/2_ in the Ni 2p spectrum in Figure 3e, respectively. This indicates that Ni exists in a +2 valence state [26]. Similarly, there is a peak at 652.27 eV in the Mn 2p_1/2_. The peak of Mn 2p_3/2_ at 641.80 eV is divided into two peaks, which demonstrates that Mn is present in the compound, similar to Mn^3+^ in Figure 3f [27].

It can be speculated that an increase in oxygen vacancy provides more active sites for electrochemical processes. At the same time, the BET-specific surface area of the CCO@NM LDH NFs is more likely to be characterized as in Figure 4. Accordingly, the specific surface area of the CCO@NM LDH NFs (~181 m^2^·g^−1^) is 3.2 times that of the CCO NFs (~56.5 m^2^·g^−1^) in Figure 4a,b. The BJH desorption average pore diameter of CCO@NM LDH NFs is estimated to be 256 nm, which is much larger than bare CCO NFs. These results indicate that CCO@NM LDH NFs possess a larger surface and abundant macropores. The large and specific surface area with a core/shell structure can offer abundant active sites for electrochemical reactions, which is convenient for the addition of more channels in the process of ion transport.

Due to its abundant oxygen vacancy defects and highly specific surface area, the CCO@NM LDH NFs were speculated as a high-performance electrode material for supercapacitors. The capacitive properties of the as-obtained CCO NF, NiMn LDH, and CCO-NM LDH NF electrodes are measured using CV, GCD, and EIS tests. Figure 5a manifests the CV plots of the CCO NF, NiMn LDH, and CCO@NM LDH NF electrodes in a potential of 0–0.8 V at a scanning rate of 10 mV·s^−1^. It is noted that the CCO@NM LDH NF electrodes show a better capacity than that of the CCO NFs and NiMn LDH electrodes. Similarly, the GCD plots of all samples at 1 A·g^−1^ are illustrated in Figure 5b. With the increase in scanning rate, the oxidation peaks in the CV curves of CCO@NM LDH NFs move to the direction of high voltage, whereas the reduction peaks move to the direction of low voltage, as shown in Figure 5c. Appendix A demonstrates the specific capacitance values of the CCO@NM LDH electrode at different scan speeds [28]. These results are attributed to the internal resistance of the electrode [29]. From the GCD curves in Figure 5d, the specific capacitance values of the CCO@NM LDH electrode at current densities of 1 A·g^−1^, 2 A·g^−1^, 4 A·g^−1^,5 A·g^−1^, 6 A·g^−1^, 8A·g^−1^, 10 A·g^−1^, and 20 A·g^−1^ are 2156.53 F·g^−1^, 2003.85 F·g^−1^, 1826.88 F·g^−1^, 1756.58 F·g^−1^, 1692.79 F·g^−1^, 1591.68 F·g^−1^, 1472.79 F·g^−1^, and 882.97 F·g^−1^. Meanwhile, the obvious voltage platform at different current densities indicates that the CCO@NM LDH electrode is made of a battery-type capacitor. The GCD curves of each electrode show excellent reversible redox behavior and pseudo-capacitance characteristics, which is consistent with the corresponding redox reaction in CV. With an increase in current density, the rate capability retention is 68.3% at a current density of 10 A·g^−1^. For comparison, CV and GCD plots of CCO NFs and NiMn LDH electrodes are given in Appendix A. We can see from Figure 5e that the discharge capacitance of the CCO@NM LDH NFs is 2156.53 F·g^−1^ at a current density of 1 A·g^−1^, which is far higher than that of the CCO (~164.51 F·g^−1^) and NiMn LDH samples (~1129.29 F·g^−1^). This is because CCO@NM LDH NFs with a large and specific surface area make better contact with the electrolyte as an active substance. The Nyquist plots of the CCO NF, NiMn LDH, and CCO@NM LDH NF electrodes are shown in Figure 5f. Meanwhile, the corresponding equivalent circuit diagram is shown in Appendix A, which consists of four major components, namely, solution resistance (R_S_), charge transfer resistance (R_CT_), double layer capacitive (C_dl_), and Warburg impedance (Zw) [30,31]. It is found that the CCO@NM LDH NF electrode possesses the smallest diffusion and charge transfer resistance, suggesting that the CCO@NM LDH NF electrode has good electronic conductivity and fast ion diffusion behaviors. After 5000 cycles, the capacity of CCO@NM LDH NFs at 5 A·g^−1^ is still 94.6%, which demonstrates the outstanding cycle stability.

The storage methods of electrode charges are divided into surface capacitance control and diffusion control [32]. The process of energy storage depends on the relationship between peak current density and the power of sweep speed: *i* = a*v*^b^. The CV curves of the CCO@NM LDH NF electrodes at a low scanning speed were tested to thoroughly understand the mechanism of charge storage inhibition in the system in Figure 6c and Appendix A, so as to ensure that ions and electrons could fully participate in the reaction. *i*_a_ represents the peak current density of the oxidation peak in the oxidation process, and *v* is the scanning rate. When b is close to 0.5, the electrochemical process mainly depends on Faradic intercalation limited by diffusion; when 0.5 < b < 1, the behavior is generally defined by the capacitance of the material in the electrode. Notably, the b value of the CCO@NM LDH NFs is the largest (0.748), reflecting that it has faster reaction kinetics in Figure 6a. Accordingly, the charge transfer process of the NiMn LDH sample is mainly controlled by diffusion, which indicates that the CCO@NM LDH NFs display battery-type behavior. In contrast, the b value of the CuCo_2_O_4_ sample as the core is 0.647. With an increase in scanning speed, the contribution of CCO@NM LDH NFs is almost entirely shown by capacitance in Figure 6d, which indicates that the charge storage controlled by the surface capacitance is dominant.

In view of its excellent capacitance performance as three electrodes, CCO@NM LDH NF, as the positive electrode, is assembled into an asymmetric capacitor. Meanwhile, activated carbon (AC) is the negative electrode, as shown in Figure 7a. The potential windows in Figure 7b of AC and CCO@NM LDH NFs come from −0.7–0 V and 0–0.8 V, respectively. Figure 7c shows the CV curves of CCO@NM LDH//AC in different voltage windows (1.2–1.8 V) at a scanning speed of 30 mV·s^−1^. When the potential window increases from 1.2 V to 1.5 V, the shape of the CV curve basically remains stable. We further confirm the rationality of the potential window through the GCD curve illustrated in Figure 7d. By synthesizing the GCD and CV curves, it is feasible to select the 0–1.5 V potential window for the assembled capacitor, i.e., CCO@NM LDH//AC. The combination of the electric double-layer capacitance and the pseudocapacitance of the storage device can be reflected by the obvious redox peak in the CV curve and the obvious potential platform in the GCD diagram [33,34]. Firstly, the CV curves (Figure 7d) of the asymmetric capacitor CCO@NM LDH//AC in the voltage window of 0–1.5 V were tested at different sweeping speeds, which displays that the asymmetric capacitor can effectively transport electrons. According to the GCD curve (Figure 7f), the discharge-specific capacitance of the asymmetric capacitor is 303.11 F·g^−1^ at a current density of 1 A·g^−1^. When the current density increases to 20 A·g^−1^, the specific capacitance of CCO@NM LDH//AC is still as high as 128 F·g^−1^, as shown in Figure 7i. With the change in current density, the coulombic efficiency is maintained at around 100%. The assembled asymmetric capacitor device provides the maximum energy density of 92.2 Wh·kg^−1^ at a power density of 730 W·kg^−1^. At the same time, the energy density is maintained at 39.55 Wh·kg^−1^ under the power density of 14,831 W·kg^−1^. This suggests that CCO@NM LDH NFs with a core-shell structure demonstrate a large specific surface area, thus providing more active sites for the electrochemical process. According to the XPS results, the NiMn LDH shell also increases abundant oxygen vacancies in CCO@NM LDH NFs, which means that more free ions and electrons can be oxidized and reduced conveniently to improve the supercapacitor’s performance. In addition, the device shows excellent stability in that its specific capacitance is 85.28% compared with that after 5000 cycles. Because of the advantages of traditional batteries and capacitors, supercapacitors not only share the high-energy density of batteries, but also express the high-power density of capacitors. In summary, the battery-type supercapacitor composed of CCO@NM LDH NFs is a promising energy storage device.

## 4. Conclusions

Dandelion-like CCO@NM LDH NFs with core/shell structure were synthesized via a two-step hydrothermal method. NiMn LDH was used as the shell to form the large and specific surface area of the CCO@NM LDH NFs (~181 m^2^·g^−1^) with more oxygen vacancy defects. Notably, the structures can provide a strong guarantee for the ions and electrons to enter and de-intercalate conveniently in the electrochemical process. As the anode material of supercapacitor, the discharge-specific capacitance is as high as 2156.53 F·g^−1^ at a current density of 1 A·g^−1^. Meanwhile, CCO@NM LDH NF electrodes also show excellent stability. When AC and CCO@NM LDH are used as the anode and cathode of the asymmetric hybrid capacitor, respectively, the device can provide a maximum energy density of 92.2 Wh·kg^−1^ at a power density of 730 W·kg^−1^. With impressive electrochemical performances, CCO@NM LDH NFs act as a battery-type electrode material on high-performance supercapacitors for energy storage.

## Data Availability

Not applicable.

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
