# Peer review of "Dandelion-Like CuCo2O4@ NiMn LDH Core/Shell Nanoflowers for Excellent Battery-Type Supercapacitor"

_nanomaterials, 2023, doi:10.3390/nano13040730_

Round 1
Reviewer 1 Report
This manuscript presents, Dandelion-like CuCo2O4@ NiMn LDH core/shell nanoflowers for excellent battery-type supercapacitor. This manuscript is well arranged and explained well. However, following revisions should be made before publication:
1. IN Figure 3 f, Mn 2p spectra is wrong.
2. The integrated area for the charge discharge curves for individual CuCo2O4 and NiMn LDH is very less compared to CuCo2O4@NiMn LDH electrode. Sufficient reason is necessary to explain this fact.
3. XRD spectrum of CuCo2O4 and NiMn LDH should be compared with CuCo2O4@NiMn LDH.
4. The HRTEM image for CuCo2O4 also should be explained.
5. The coulombic efficiency of the CuCo2O4@NiMn LDH electrode should be provided.
6. In the GCD curves of the asymmetric device, In figure 7 d, the device has different GCD curves that in Figure 7 f, Why?
7. Why potential window for CV and GCD in three electrode system is different? The CV and GCD curves clearly depict battery type electrode materials. Therefore, specific capacity calculation is recommended. Refer: doi.org/10.1016/j.cej.2021.132345, doi.org/10.1016/j.compositesb.2022.110339.
8. EIS circuit diagram should be updated and corresponding parameter can be explained with the reference of following article: doi.org/10.1016/j.cplett.2022.139884.
Author Response
This manuscript presents, Dandelion-like CuCo2O4@ NiMn LDH core/shell nanoflowers for excellent battery-type supercapacitor. This manuscript is well arranged and explained well. However, following revisions should be made before publication:
- IN Figure 3 f, Mn 2p spectra is wrong.
Thank you very much for pointing out this obvious mistake. We have modified Mn 2p spectra of the corresponding XPS diagram in Figure 3 f and marked it in red in manuscript.
- The integrated area for the charge discharge curves for individual CuCo2O4 and NiMn LDH is very less compared to CuCo2O4@NiMn LDH electrode. Sufficient reason is necessary to explain this fact.
Thanks for the questions raised by the reviewers. By combining XPS and bet results, we know that a large number of oxygen vacancies are provided due to the synthesis of core-shell structure. More importantly, the specific surface area of core-shell structure is much higher than that of individual CuCo2O4 and NiMn LDH, which provides strong evidence for its excellent performance in electrochemical process.
- XRD spectrum of CuCo2O4 and NiMn LDH should be compared with CuCo2O4@NiMn LDH.
Thanks to the reviewers for the valuable comment. Figure 2(g) shows the XRD pattern of CuCo2O4 and CuCo2O4-NiMn LDH samples.Then, the red diffraction peaks of LDH can be reasonably indexed to a series of crystal planes, such as (003), (006), (009) and (110).
- The HRTEM image for CuCo2O4 also should be explained.
Thank you very much for asking this important question. Due to the Spring Festival holiday, the instrument characterization can't work normally for the time being, so we can't provide the corresponding the HRTEM image for CuCo2O4. Time is pressing, I'm very sorry.
- The coulombic efficiency of the CuCo2O4@NiMn LDH electrode should be provided.
Thank the reviewers for their suggestions. The coulombic efficiency of the CuCo2O4@NiMn LDH electrode is shown in the Figure 7(i).
- In the GCD curves of the asymmetric device, In figure 7 d, the device has different GCD curves that in Figure 7 f, Why?
Thanks to the reviewers for the valuable question. Fig. 7d is a window without voltage platform selected under different voltage windows. Then, under this voltage window (0~1.5V), Figure 7f shows the GCD diagram under different current densities.
- Why potential window for CV and GCD in three electrode system is different? The CV and GCD curves clearly depict battery type electrode materials. Therefore, specific capacity calculation is recommended. Refer: doi.org/10.1016/j.cej.2021.132345, doi.org/10.1016/j.compositesb.2022.110339.
Thank you very much for your valuable comments. The potential window of GCD with three electrodes in the manuscript is 0 ~ 0.6V, which is smaller than that of CV curve (0 ~ 0.8V). This is mainly because in the charging process, the large voltage window can not be reached, so the small window is selected. The corresponding capacitance calculation of CV curves at different scanning speeds is supplemented in Figure S5. Corresponding references have been cited in corresponding positions in this paper.
- EIS circuit diagram should be updated and corresponding parameter can be explained with the reference of following article: doi.org/10.1016/j.cplett.2022.139884.
Thanks to the reviewers for the suggestion. EIS circuit diagram is added in Figure S6, the corresponding parameter can be explained with the reference.
Reviewer 2 Report
In this manuscript, the authors describe the synthesis and characterization of CuCo2O4-NiMn layered double hydroxide composites in which the hydroxide constitutes the shell. These nanocomposites have a flower-like morphology, which ensures a high surface area, desirable for supercapacitor applications. What is the reason for adding a CoCl2 solution in the second step of the synthesis, along with hexamethylenetetramine and NiCl2? The spinel [hase has been obtained already after the first step. How reproducible the synthesis is and what is the typical amount of sample that can be prepared in a single batch? Stating that the nickel foam was used as a substrate is inappropriate since the foam serves as support, not substrate. The authors claim that the NIMn layered oxide shells are ultrathin. How thin are they? What is the chemical composition of these NiMn shells? What do the Ov and Ow stand for? Ow apparently are absorbed water molecules, this should be clearly stated in the manuscript. The authors claim that the samples possess "abundant oxygen vacancy defects" (Page 6) but no numerical values are provided to support this claim. The electrochemical characterization of the samples is adequate, some minor English corrections are needed. Based upon the foregoing, I consider that this manuscript could be worth publishing after the authors address the comments and suggestions above.
Author Response
Thank you very much for your valuable comments. Firstly, the 0.198 g of MnCl2·4H2O is instead of CoCl2. Adding a CoCl2 solution in the second step of the synthesis, along with hexamethylenetetramine and NiCl2 can synthesize the NiMn LDH. We had made a mistake in the second step of the synthesis. The repeatability of the sample is high. The mass of CCO@NM LDH samples loaded on NF is about 1.2 g. The typical amount of CCO@NM LDH sample that can be prepared in a single batch is about 0.4 g. Thus, the Ni foam in the experiment serves as support. From the TEM image, the NiMn LDH sample has good light transmittance, which demonstrates that the LDH sample is very thin. Compared with the AFM data of LDH samples prepared before, the thickness of NiMn LDH is about 5 nm. NiMn LDH can be reasonably indexed to a series of crystal planes, such as (003), (006), (009) and (110). It can be seen that the appearance of OW located at 532.4 eV induced physically and chemically absorbed water molecules on the surface of CCO@NM LDH NFs. Meanwhile, Ov stands for oxygen vacancy. Based on the characterization results of XPS, the area of Ov peak of CCO@NM LDH NFs marked green is larger than that of CCO. Thus, the CCO@NM LDH NFs sample possess abundant oxygen vacancy defects for enhancing the performance.
Round 2
Reviewer 1 Report
Authors responds reviewers comment. This manuscript can be accept for publication.